# Decision-Making in Repeated Games: Insights from Active Inference

**DOI:** 10.3390/bs15121727

**Published:** 2025-12-13

**Authors:** Hui Yuan, Ligang Wang, Wenbin Gao, Ting Tao, Chunlei Fan

**Affiliations:** 1State Key Laboratory of Cognitive Science and Mental Health, Institute of Psychology, Chinese Academy of Sciences, Beijing 100101, China; yuanh@psych.ac.cn (H.Y.); gaowb@psych.ac.cn (W.G.); taot@psych.ac.cn (T.T.); fancl@psych.ac.cn (C.F.); 2Department of Psychology, University of Chinese Academy of Sciences, Beijing 100049, China

**Keywords:** repeated games, decision-making, computational modeling, active inference

## Abstract

This review systematically explores the potential of the active inference framework in illuminating the cognitive mechanisms of decision-making in repeated games. Repeated games, characterized by multi-round interactions and social uncertainty, closely resemble real-world social scenarios in which the decision-making process involves interconnected cognitive components such as inference, policy selection, and learning. Unlike traditional reinforcement learning models, active inference, grounded in the principle of free energy minimization, unifies perception, learning, planning, and action within a single generative model. Belief updating occurs by minimizing variational free energy, while the exploration–exploitation dilemma is balanced by minimizing expected free energy. Based on partially observable Markov decision processes, the framework naturally incorporates social uncertainty, and its hierarchical structure allows for simulating mentalizing processes, providing a unified account of social decision-making. Future research can further validate its effectiveness through model simulations and behavioral fitting.

## 1. Introduction

Decision-making in repeated games is a crucial aspect of human intelligence and has been the focus of extensive research across fields such as psychology and artificial intelligence ([2]; [76]). In repeated games, participants engage in multiple rounds of interpersonal interactions that closely resemble the real social world—a world frequently portrayed as more unpredictable and uncertain than the non-social world ([18]). Decision-making within social contexts presents significant complexity and difficulty. This stems not only from the challenge of identifying choices that maximize self-interest given the unpredictability of others’ behaviors, but also from the necessity of reconciling the self-interest with the interests of others, which requires a trade-off between cooperation and competition ([42]). Such decision-making presents significant challenges for both artificial intelligence and human agents. Consequently, a deeper investigation into its cognitive mechanisms is crucial not only for enhancing human decision-making capabilities but also for advancing AI towards greater intelligence, enhanced adaptability to complex social environments, and more effective collaboration with humans. 

Traditional psychological methods are limited in their ability to assess such complex dynamic processes. Computational modeling, as a quantitative methodology characterized by rigor, scientific precision, and interpretability, has been extensively applied in domains such as computational psychiatry ([32]; [48]), and has demonstrated emerging utility in social psychology ([15]; [28]). It enables the simulation of complex cognitive processes underlying behavioral phenomena through mathematical formalisms, provides rigorous scientific characterization of the dynamic mechanisms governing human behavior, and derives latent variables from behavioral data that are not directly observable ([47]). In doing so, it establishes a novel theoretical framework for advancing understanding of psychological and neural mechanisms, while overcoming the limitations of traditional research methods. 

Various families of computational models have been developed to characterize the decision-making process. Unlike reinforcement learning, which frames decision-making as an adaptive behavior driven by the principle of maximizing external rewards ([73]), active inference conceptualizes decisions as a process of minimizing free energy ([20]). In this framework, the decision-making agent explores the environment to reduce uncertainty, minimizing the discrepancy between anticipated outcomes and preferred outcomes, while dynamically integrating perception and policy selection. The active inference framework offers a promising model for simulating human cognitive processes in partially observable environments, such as social contexts. This approach could significantly deepen our understanding of how human intelligence navigates adaptive decision-making within ever-changing social settings.

## 2. Game Theory and Repeated Games

### 2.1. The History of Game Theory Development

The development of game theory has undergone a long process from ideas to theory, and from simplicity to complexity, as shown in Figure 1. Long before the formal establishment of modern game theory, game-theoretic ideas were already present. As early as in ancient times, Sun Tzu’s *The Art of War* in China and Machiavelli’s *The Prince* in the West contained the core game theory concepts, such as strategic interaction and the balancing of interests. In the 16th century, the Italian mathematician Girolamo Cardano, in his book *Liber de Ludo Aleae*, was the first to apply mathematical methods to analyze the probabilities and payoffs of gambling games like dice, marking the germination of game theory thought.

The work of John von Neumann established game theory as an independent field in the first half of the 20th century. In 1928, Von Neumann published *On the Theory of Parlor Games*, proving for the first time the minimax theorem, which provided a solid mathematical foundation for two-person zero-sum games ([50]). He simplified its proof as an extension of Brouwer’s fixed-point theorem, a method that later became standard in game theory and mathematical economics. Collaborating with economist Oskar Morgenstern, von Neumann developed these ideas into a monumental work published in 1944, *Theory of Games and Economic Behavior*. This book was the first to establish a systematic axiomatic framework for game theory; it explicitly positioned game theory as an analytical tool for economics, laying the foundation for its later widespread application ([78]).

After game theory became an independent discipline, its core concepts were continuously broken through and refined, leading to a significant improvement of its theoretical system. John Nash made outstanding contributions by proposing the most important concept in game theory—the Nash Equilibrium ([49]). In a multi-player game, when the strategies of all players form a stable combination such that no player can improve their own payoff by unilaterally changing their own strategy, this strategy profile constitutes a Nash Equilibrium. The Nash Equilibrium is a more general solution concept than the traditional minimax solution, applicable not only to zero-sum games but to all game models. John Harsanyi proposed the Bayesian Nash Equilibrium, applicable to situations where players do not have complete information about their opponents’ types ([31]). By introducing a probability distribution over the opponent’s types, he successfully transformed incomplete information into imperfect information, breaking through the previous limitation of game theory focusing mostly on complete information. This allowed for a more accurate simulation of human decision-making under uncertainty. Reinhard Selten proposed the Subgame Perfect Nash Equilibrium, which ensures that equilibrium strategies are reasonable and feasible in every subgame by eliminating incredible threats in sequential games, thus refining the Nash equilibrium from a dynamic perspective ([63]). For constructing a complete, rigorous, and realistic theoretical framework for game theory and making groundbreaking contributions to equilibrium analysis, John Nash, John Harsanyi, and Reinhard Selten jointly received the Nobel Memorial Prize in Economic Sciences in 1994.

In recent years, the development of game theory has gradually extended beyond the boundaries of economics, deeply integrating with psychology, biology, computer science, and other fields, leading to a series of interdisciplinary studies. Scholars like Daniel Kahneman introduced cognitive biases into game analysis, revising the assumption of perfect rationality and promoting the development of behavioral game theory ([36]). John Maynard Smith, among others, pioneered evolutionary game theory, using the concept of an Evolutionarily Stable Strategy to explain competition and cooperation in animal behavior ([64]). In the field of computer science, game theory provides a framework for strategy optimization and equilibrium decision-making in multi-agent interactive scenarios, while computer science, in turn, provides efficient algorithms for solving game-theoretic problems ([52]).

### 2.2. Repeated Games

Based on the sequence of players’ actions and the availability of information, games can be classified into static games and dynamic games. In static games, players act simultaneously with no knowledge of prior actions; whereas in dynamic games, players move sequentially and have access to the observable history of play. Repeated games are a special form of dynamic game, where a stage game with the same structure is played multiple times in succession. The conditions, rules, and content of each iteration remain identical, and participants can observe historical behaviors after each round of interaction, requiring them to balance short-term gains with long-term interests when making decisions.

Previously introduced scholars John Nash and Reinhard Selten made significant contributions to establishing equilibrium concepts in repeated games. Beyond their work, many other researchers have conducted in-depth studies on repeated games. Robert Aumann pioneered the theory of infinitely repeated games, demonstrating that rational players can achieve stable cooperation through trigger strategies (e.g., continuing to cooperate as long as the opponent cooperates, but triggering retaliatory defection upon observing the opponent’s betrayal). This resolved the core puzzle of how cooperation can emerge in repeated games and provided a theoretical framework for long-term interactive scenarios ([4]). Thomas Schelling integrated repeated games with real-world social contexts, refining the classification framework into conflict games, cooperative games, and coordination games. He also introduced concepts such as focal points and commitment strategies, explaining how participants in repeated interactions can quickly reach cooperation through shared cognition ([62]). In the Prisoner’s Dilemma tournament, researchers verified the optimality of the tit-for-tat strategy (i.e., imitating the opponent’s move from the previous round in each subsequent round) in repeated games ([5]). This strategy, being simple and clear, consistently promoted cooperation and provided strong empirical and simulation-based support for understanding cooperative behavior in the real world.

Depending on the nature of interdependence among players’ interests, repeated games can be categorized into three distinct types: conflict games, cooperation games, and coordination games ([62]).

Conflict games, also known as zero-sum games, are strategic situations in which the interests of players are inherently antagonistic. In these games, the total sum of payoffs and losses across all players always equals zero, meaning one player’s gain necessarily implies another’s loss ([62]). Within this framework, cooperation among players is structurally impossible; instead, decision-making is premised on competition. Participants focus on concealing their own strategic intentions while attempting to infer opponents’ strategies and psychological states to optimize their own outcomes. Conflict games manifest ubiquitously across real-world scenarios, encompassing competitive sports (e.g., table tennis), board games (e.g., chess), as well as the game of rock-paper-scissors. In psychological research, such adversarial frameworks are frequently used as experimental paradigms to investigate deception, inequality, and related strategic performance ([41]; [85]).

Cooperation games, also known as mixed-motive games, characterize interactions where individuals face both competitive and collaborative incentives, resulting in payoff structures that are neither purely adversarial nor fully aligned. These games inherently involve resource allocation between self and others, compelling players to navigate a critical trade-off: whether to cooperate to maximize collective benefits or defect to secure personal gains ([46]). Unlike zero-sum conflict games, participants in cooperative frameworks exhibit stronger motivations to understand opponents’ strategies ([81]). Cooperation games such as the Prisoner’s dilemma, trust game, and dictator game are widely employed to investigate the formation of cooperation and defection ([7]; [43]; [56]). The Prisoner’s dilemma stands as a canonical example, in which two players independently choose between “cooperation” and “defection” without communication. Its payoff structure is characterized by four critical outcomes: mutual cooperation yields moderate rewards for both players; mutual defection results in moderate penalties; while unilateral defection provides the defector with maximum gains at the expense of the cooperator, who incurs the highest penalty. This matrix is shown in Figure 2. The payoff matrix of the game describes the strategies available to players and their corresponding outcomes. Based on this matrix, strategies and their expected utilities can be quantitatively analyzed. In the one-shot Prisoner’s dilemma, defection constitutes the dominant strategy that maximizes individual payoff under rationality. However, in the repeated game, persistent defection risks triggering retaliatory defection from opponents in subsequent rounds, reducing long-term cumulative gains. It is within this dynamic that cooperation gradually forms ([11]). 

The coordination game is a type of game in which the interests of participants are consistently aligned. In such a game, the payoffs and losses for all participants are identical, meaning that both parties will either succeed or fail together ([62]). Under the rules of this game, due to the congruence of interests between the two sides, cooperation becomes their inevitable choice. The players only need to consider how to coordinate with each other, rather than deliberating on how to allocate resources or resolve conflicts of interest. Coordination games, such as matching games (e.g., the Heads-Tails game), involve scenarios in which players receive a positive payoff only if they choose the same option, whereas mismatched choices result in no payoff for either. Although less frequently studied in decision-making research, these games are commonly used to investigate inter-brain synchronization during cooperation ([14]; [53]).

## 3. Decision-Making in Repeated Games

Within social environments, human decision-making requires explicit consideration of others’ behaviors. Games provide a valuable methodological framework for quantitatively assessing such social decisions, with structured interaction scenarios enabling the study of social cognitive processes such as cooperation, competition, and trust formation ([45]; [51]; [85]). Unlike single-round games in which participants have no shared history or future, repeated games more closely resemble real-life social interactions. These interactions involve multiple sequential encounters, with participants receiving feedback between rounds. The feedback enables participants to update their beliefs about the opponent and adjust their strategy accordingly, resulting in dynamically evolving behavior over time ([9]).

On the spectrum of games, pure coordination games and zero-sum games represent the extremes, while mixed-motive games, which incorporate elements of both cooperation and conflict, occupy the intermediate space ([62]). However, in the context of repeated games, regardless of the players’ incentive structures, the opponent’s behavior directly impacts the decision-maker’s choices and outcomes. The opponent’s hidden states and actions introduce social uncertainty that extends beyond non-social uncertainty. Social uncertainty refers to a person’s inability to precisely predict their own future states and actions due to uncertainty about others’ states and actions ([18]). The repeated game continuum provides a powerful framework for investigating decision-making processes under such social uncertainty.

Extensive research has focused on decision-making under non-social uncertainty. Individuals prioritize attention to cues that can reduce uncertainty ([80]) and pay greater attention to options with higher uncertainty ([71]). Increased uncertainty enhances individuals’ learning rates ([70]) and promotes exploratory behaviors ([72]). While the mechanisms by which non-social uncertainty influences decision-making can be extended to social uncertainty to some extent, social contexts inherently involve greater unpredictability. This is because the behavioral motives of others are difficult to directly observe and evolve dynamically over time. Additionally, in contexts involving repeated interactions with others—such as repeated games—suboptimal decisions can have long-lasting negative effects. These effects may hinder the formation of cooperation with others and ultimately compromise one’s own interests.

To resolve social uncertainty and make adaptive decisions, individuals develop specialized mechanisms in which social inference plays an important role. Inference can be categorized into automatic and controlled inference based on the degree of cognitive control and the corresponding effort costs ([18]). When encountering opponents in a game, individuals rapidly and automatically form initial impressions—including whether they are trustworthy, threatening or risk-seeking—based on the opponent’s features like skin color and clothing ([34]). When information about others is scarce, people quickly resort to established social norms, assuming others are more likely to make decisions based on principles like cooperation and trust ([19]). Automatic inference consumes minimal cognitive resources and can strongly constrain subsequent predictions about opponents. In contrast, controlled inference requires greater cognitive control to infer the opponent’s motives and specific behaviors in a more nuanced manner, thereby updating the initial impression. People often engage in perspective-taking, adopting the viewpoint of others or their situational context to imagine or infer their perspectives and attitudes ([17]; [25]). This process narrows the scope of predicting others’ behaviors and further reduces social uncertainty. 

Based on inferences about their opponent, individuals need to engage in appropriate policy selection in games. Traditional behavioral game theory posits that policy selection is based on the principle of expected utility maximization ([78]). Individuals are assumed to rationally weigh the future rewards of different action sequences and choose the policy that yields the highest payoff. However, this principle faces explanatory limitations in social decision-making contexts characterized by high uncertainty ([10]). It relies on the individual’s ability to form and update precise probabilistic beliefs about others’ behavior, which poses a significant challenge when the opponent’s strategy is unknown or dynamically changing. The recently developed active inference framework offers a novel perspective. It proposes that policy selection is grounded in the principle of free energy minimization—that is, minimizing prediction error ([55]). This process entails both minimizing the discrepancy between actual outcomes and the individual’s preferred outcomes, and actively reducing uncertainty about the environment. Consequently, the driver of policy selection expands from merely pursuing reward to a dual pursuit of both pragmatic and epistemic value.

In repeated games, the outcomes resulting from policy selections directly guide an individual’s subsequent decisions. Participants can directly observe opponents’ responses to their behavioral choices, receive feedback between game rounds, and engage in learning ([6]). They integrate new feedback evidence with prior inferences by weighting these information sources, thereby updating their predictions about the opponent ([70]). Depending on whether the feedback is consistent with prior predictions, it can broaden or narrow the belief distribution concerning the opponent’s likely behaviors ([18]). As the game unfolds, participants continuously gather information about the opponent, learning their behavioral patterns. The learning rate is influenced by uncertainty; at the beginning of repeated games, when social uncertainty is highest, the learning rate is also at its peak ([13]; [70]). At the same time, participants face a critical exploration–exploitation trade-off during learning ([26]; [40]). They must weigh the choice between exploring unknown strategies to observe the opponent’s response and gain information about the value of such behaviors, versus exploiting known strategies that maximize immediate payoff based on past experience ([69]). Excessive exploration may lead to reduced efficiency, while prematurely exploiting existing experience may result in missing out on better strategies.

Social gaming situations evoke high levels of uncertainty, during which the decision-making process, as illustrated in Figure 3, involves interrelated mechanisms of inference, policy selection, and learning. Automatic inference and controlled inference operate simultaneously as two poles of a continuum, constraining predictions regarding the intentions and behaviors of opponents. These predictions guide participants’ strategic choices based on different principles and are continuously updated through learning in repeated games, thereby further reducing social uncertainty. Human social cognition resembles a sophisticated computational system, allowing individuals to predict, respond to, and coordinate with others’ behaviors, thereby underpinning the stability of social interactions.

## 4. Computational Modeling for Decision-Making

With the advancement of computational science, computational ideas pervade many areas of science and play an integrative explanatory role in cognitive sciences and neurosciences ([48]). While focusing on human social behavior, social psychology is committed to explaining social life based on the general principles of human cognition ([16]). Computational ideas have thus been increasingly adopted and utilized by social psychology researchers. Computational modeling offers a rigorous framework for describing abstract theories underlying human sociality in a clear and interpretable manner ([15]). Since decision-making in repeated games involves recursive cognitive processes such as inference and learning—where recursivity, characterized by cyclic causal relationships, represents a core concept in computational modeling ([60]). Additionally, computational modeling establishes a system of interacting relationships, allowing models of inference and learning to be integrated to capture more complex cognitive processes. Thus, employing computational modeling to investigate the underlying social psychological mechanisms in repeated games is highly feasible.

Current computational models for decision-making research can be primarily categorized into two major classes: reinforcement learning models and Bayesian models. Reinforcement learning models, grounded in the rational agent assumption, are widely used in the field to simulate how individuals learn from outcome feedback during interactive behaviors ([74]; [75]; [84]). These models conceptualize the individual-environment interaction as a Markov Decision Process (MDP), in which the agent can observe all possible states of the environment and influence it through actions. As the state of the environment transitions, a reward function provides feedback to the individual, who in turn adapts their behavior based on the principle of maximizing cumulative rewards ([59]). Researchers have integrated various cognitive strategies with reinforcement learning frameworks to develop a range of decision-making models. For instance, the Prospect-Valence Learning model posits that individuals evaluate the expected utility of different options and update expected valences using reinforcement learning rules to guide decisions ([1]). Similarly, the Valence-Plus-Perseverance model combines heuristic strategies with reinforcement learning, proposing that choices are influenced by previous actions and their outcomes, which are integrated with expected valence to inform subsequent decision-making ([83]).

The rational agent assumption of reinforcement learning models is not fully consistent with individuals’ decision-making behaviors in daily life, and the environment in which individuals operate is not fully observable—instead, it is fraught with uncertainty. Therefore, unlike reinforcement learning models, Bayesian models for decision-making are based on the Partially Observable Markov Decision Process (POMDP), which formalizes the state of the decision-making environment as only partially observable ([35]). Bayesian models assume that individuals exhibit bounded rationality and hold their own prior beliefs about environmental states; they can update these beliefs based on feedback obtained from interactions to form posterior beliefs, and then make decisions accordingly. In comparison, the Bayesian framework conceptualizes decision-making as a process of belief updating based on both internal and external information under uncertainty, thereby offering a more reasonable explanation for decision-making behavior in everyday social interactions ([18]). Meanwhile, the Bayesian framework does not require individuals to behave in a strictly Bayes-optimal manner. Rather, approximate Bayesian inference processes can effectively account for human decision-making behavior.

## 5. Basic Concepts of Active Inference

Active inference, as a specific implementation and computational framework of approximate Bayesian inference, is founded on a key premise: that organisms possess an internal generative model of the statistical regularities of their environment. Equipped with this model, an organism can infer the hidden causes of its perception and select optimal actions to achieve desired outcomes. Simultaneously, the active inference framework is underpinned by two core concepts ([65]). The first is that decision-makers are not simply passive Bayesian observers. Instead, they actively engage with the environment to gather information and seek preferred observations. The second is Bayesian inference, which refers to the uncertainty-weighted updating of prior beliefs based on the observed distribution of possible outcomes, as formalized by Bayes’ theorem:(1)$$p\left(s|o\right)\;=\;\frac{p\left(o|s\right)p\left(s\right)}{p\left(o\right)}$$

In Bayes’ theorem, the left-hand side, *p*(*s*|*o*), represents the posterior belief about possible states (*s*)—that is, the updated belief distribution after incorporating a new observation (*o*). Here, *s* is an abstract variable that can represent any entity about which beliefs can be formed. On the right-hand side, *p*(*s*) denotes the prior belief about *s* before acquiring the new observation. The term *p*(*o*|*s*) is the likelihood term, representing the probability of observing *o* given that the state is *s*. The denominator *p*(*o*) represents the model evidence, also known as the marginal likelihood, which indicates the total probability of observing *o*. It serves as a normalization constant, ensuring that the posterior belief constitutes a valid probability distribution.

For instance, in a cooperation game, we need to infer whether an opponent is more inclined to cooperate or defect. Here, *s* represents the opponent’s behavioral type. Assuming the opponent can be coarsely categorized into two types: *s*_1_ represents a cooperative type and *s*_2_ represents a defecting type. The variable *o* corresponds to the observed actual behavior of the opponent in each round of the game. Prior to the game, we can form an initial impression of the opponent through social inference. For example, if we believe the opponent is inclined to cooperate, we may assign a higher prior probability to *s*_1_, such as *p*(*s*_1_) = 0.8, and consequently *p*(*s*_2_) = 0.2. Suppose during the feedback phase of the game, we observe our opponent’s defection. In this context, *p*(*o*|*s*_1_) signifies the probability that a cooperative-type opponent would defect; this value is expected to be low, for instance, *p*(*o*|*s*_1_) = 0.3. Conversely, *p*(*o*|*s*_2_) denotes the probability that a defecting-type opponent would defect, which would be higher, e.g., *p*(*o*|*s*_2_) = 0.7. The marginal likelihood *p*(*o*), which acts as a normalizing constant, is calculated as the total probability of observing a defection across all possible opponent types: *p*(*o*) = *p*(*o*|*s*_1_) × *p*(*s*_1_) + *p*(*o*|*s*_2_) × *p*(*s*_2_) = (0.3 × 0.8) + (0.7 × 0.2) = 0.38. The posterior belief that the opponent is cooperative is then updated via Bayes’ theorem: *p*(*s*_1_|*o*) = [*p*(*o*|*s*_1_) × *p*(*s*_1_)]/*p*(*o*) = (0.3 × 0.8)/0.38 ≈ 0.632. Similarly, the posterior belief that the opponent is defecting is: *p*(*s*_2_|*o*) = [*p*(*o*|*s*_2_) × *p*(*s*_2_)]/*p*(*o*) = (0.7 × 0.2)/0.38 ≈ 0.368. Based on this Bayesian update, after observing one instance of defection, the player’s belief about the opponent’s behavioral pattern shifts: the probability assigned to the cooperative type decreases from 80% to approximately 63.2%, while the probability of the defecting type increases from 20% to 36.8%. In subsequent rounds of the game, this posterior belief becomes the new prior. The player continues to perform Bayesian updates based on the opponent’s new actions, iteratively refining their beliefs to better approximate the opponent’s true behavioral pattern.

In this simple example, we were able to easily compute Bayes’ theorem numerically. However, beyond the simplest belief distributions, the marginal likelihood *p*(*o*) in Bayes’ theorem is computationally intractable. It requires summing the probability of observations under all possible states. As the number of state dimensions increases, the number of terms to sum grows exponentially. For instance, in a game, an opponent’s type might be defined by multiple parameters such as cooperative tendency, risk aversion, and capability. In such cases, calculating *p*(*o*) becomes an integration over an ultra-high-dimensional space, which is infeasible to perform directly. Since computing posterior beliefs through exact Bayesian inference is infeasible in complex models, approximation techniques are required to solve this problem. This computational infeasibility is the fundamental reason for the central concept of variational free energy (*VFE*) in active inference.

Since the exact posterior distribution *p*(*s*|*o*) is computationally intractable, a simple approximate posterior distribution *q*(*s*) is introduced. By adjusting parameters through optimization algorithms, *q*(*s*) is made to approximate *p*(*s*|*o*) as closely as possible, transforming the uncomputable Bayesian inference into an optimization problem of minimizing a computable function. The metric used to quantify the discrepancy between two distributions is the Kullback–Leibler (KL) Divergence. The closer two distributions match, the smaller the KL divergence. Through mathematical derivation based on the relevant equation, the divergence between *q*(*s*) and *p*(*s*|*o*) can be expressed as follows:(2)$$\begin{array}{ll} D_{\mathrm{KL}}\left[q\left(s\right)∥p\left(s|o\right)\right] & =\sum_{s}q\left(s\right)\ln\frac{q\left(s\right)}{p\left(s|o\right)}\\ & =\sum_{s}q\left(s\right)\left[\ln q\left(s\right)-\ln p\left(s|o\right)\right]\\ & =\sum_{s}q\left(s\right)\left[\ln q\left(s\right)-\ln\frac{p\left(o,s\right)}{p\left(o\right)}\right]\\ & =\sum_{s}q\left(s\right)\left[\ln q\left(s\right)-\ln p\left(o,s\right)+\ln p\left(o\right)\right]\\ & =\sum_{s}q\left(s\right)\ln\frac{q\left(s\right)}{p\left(o,s\right)}+\ln p\left(o\right)\\ & =E_{q\left(s\right)}\left[\ln\frac{q\left(s\right)}{p\left(o,s\right)}\right]+\ln p\left(o\right) \end{array}$$

According to Equation (2), in order to make *q*(*s*) as close as possible to *p*(*s*|*o*), it is necessary to minimize $D_{\mathrm{KL}}\left[q\left(s\right)∥p\left(s|o\right)\right]$. Since ln *p*(*o*) is a constant term independent of *q*(*s*), minimizing the KL divergence is equivalent to minimizing $E_{q\left(s\right)}\left[\ln\frac{q\left(s\right)}{p\left(o,s\right)}\right]$, which is defined as the variational free energy (*F*). Since $D_{\mathrm{KL}}\left[q\left(s\right)∥p\left(s|o\right)\right]$ ≥ 0, it can be inferred through equation transformation that the variational free energy *F* ≥ $-\ln p\left(o\right)$. $-\ln p\left(o\right)$ represents the surprise of sensory input, where $p\left(o\right)$ denotes the overall probability of obtaining an observation, also known as model evidence. A lower value of $p\left(o\right)$ corresponds to a higher degree of surprise in the sensory input. For example, the probability of seeing an elephant in an office is very low, so the surprise associated with such an observation is high. As the variational free energy (*F*) is an upper bound on surprise, minimizing the variational free energy consequently minimizes surprise indirectly.

The equation for variational free energy and its further derivation are shown below. It is worth noting that in the active inference framework, variational free energy is typically computed conditioned on different policies (*π*). This is because active inference emphasizes that agents actively infer future observations based on potential courses of action, hence the approximate posterior and other quantities are conditioned on policies.(3)$$\begin{array}{ll} F_{\pi } & =E_{q\left(s|\pi \right)}\left[\ln\frac{q\left(s|\pi \right)}{p\left(o,s|\pi \right)}\right]\\ & =E_{q\left(s|\pi \right)}\left[\ln q\left(s|\pi \right)-\ln p\left(o,s|\pi \right)\right]\\ & =E_{q\left(s|\pi \right)}\left[\ln q\left(s|\pi \right)-\ln p\left(s|\pi \right)\right]-E_{q\left(s|\pi \right)}\left[\ln p\left(o|s,\pi \right)\right]\\ & =D_{\mathrm{KL}}\left[q\left(s|\pi \right)∥p\left(s|\pi \right)\right]-E_{q\left(s|\pi \right)}\left[\ln p\left(o|s\right)\right] \end{array}$$

According to the last line of Equation (3), minimizing variational free energy is equivalent to minimizing $D_{\mathrm{KL}}\left[q\left(s|\pi \right)∥p\left(s|\pi \right)\right]$ while maximizing $E_{q\left(s|\pi \right)}\left[\ln p\left(o|s\right)\right]$. $D_{\mathrm{KL}}\left[q\left(s|\pi \right)∥p\left(s|\pi \right)\right]$ refers to the difference between the approximate posterior distribution *q*(*s*|*π*) and the prior distribution *p*(*s*|*π*) under a given policy *π*. It represents the cost incurred to align posterior beliefs with prior expectations, serving as a measure of complexity. Minimizing this term means the agent tends to choose policies that minimize changes in its beliefs. Excessively high complexity leads to a phenomenon analogous to overfitting in statistics, where the agent adjusts its model to accommodate the randomness of a particular observation, thereby reducing its overall predictive capability. $E_{q\left(s|\pi \right)}\left[\ln p\left(o|s\right)\right]$ is the expected log-likelihood, i.e., the expected logarithmic probability of observing *o* given state *s*. It reflects how accurately the model predicts observations based on beliefs. Therefore, minimizing variational free energy is achieved by simultaneously minimizing model complexity and maximizing predictive accuracy.

The active inference framework treats perception and learning as processes of minimizing variational free energy ([22]). Perception corresponds to updating posterior beliefs in real-time based on each new observation, providing the best interpretation of sensory input; learning corresponds to gradually adjusting model parameters over long-term observations to align with accumulated experience. In the processes of perception and learning, the agent is not solely concerned with finding the best-fitting posterior. It also strives to update its beliefs in the most parsimonious way, avoiding excessive deviations from prior beliefs, thereby achieving a balance between accuracy and complexity.

The active inference addresses not only the processing of past and present information but also encompasses the planning and action selection concerning future states. Similar to the principles underlying perception and learning, the goal of planning and action selection is to choose a policy *π* that minimizes variational free energy in the future. The key distinction lies in the extension of the variational free energy to incorporate expected future observations, yielding the expected free energy (*EFE*). The specific derivation of the expected free energy (*G*) is shown below. The first two lines of its equation closely resemble those of the variational free energy, with the only difference being the inclusion of future observations *o* under expectation. After decomposing expected free energy into two components related to information seeking and reward seeking in the third line, the fourth line introduces the agent’s preferences *C*. As a decision-making model, active inference similarly requires encoding preferences. Unlike reinforcement learning, which encodes preferences as an external reward function, active inference internalizes them as part of the generative model by incorporating preferred observations $p\left(o|C\right)$ into the expected free energy.(4)$$\begin{array}{ll} G_{\pi }= & E_{q\left(o,s|\pi \right)}\left[\ln\frac{q\left(s|\pi \right)}{p\left(o,s|\pi \right)}\right]\\ & =E_{q\left(o,s|\pi \right)}\left[\ln q\left(s|\pi \right)-\ln p\left(o,s|\pi \right)\right]\\ & =E_{q\left(o,s|\pi \right)}\left[\ln q\left(s|\pi \right)-\ln p\left(s|o,\pi \right)\right]-E_{q\left(o|\pi \right)}\left[\ln p\left(o|\pi \right)\right]\\ & \approx E_{q\left(o,s|\pi \right)}\left[\ln q\left(s|\pi \right)-\ln q\left(s|o,\pi \right)\right]-E_{q\left(o|\pi \right)}\left[\ln p\left(o|C\right)\right]\\ & ={-E}_{q\left(o,s|\pi \right)}\left[\ln q\left(s|o,\pi \right)-\ln q\left(s|\pi \right)\right]-E_{q\left(o|\pi \right)}\left[\ln p\left(o|C\right)\right] \end{array}$$

In Equation (4), the first term $E_{q\left(o,s|\pi \right)}\left[\ln q\left(s|o,\pi \right)-\ln q\left(s|\pi \right)\right]$ in the final line represents the epistemic value, which expresses the expected log difference between the approximate posterior (based on the policy and the observations it generates) and the prior (based on the policy). A greater difference between the prior and the posterior indicates that the agent acquires more new information, implying a greater information gain from the policy and, consequently, a higher epistemic value. According to the equation, a higher epistemic value leads to a lower expected free energy for that policy. The second term $E_{q\left(o|\pi \right)}\left[\ln p\left(o|C\right)\right]$ represents the pragmatic value, which expresses the expected log probability of the agent’s preferred observations. A higher probability of preferred outcomes indicates that the agent is more likely to obtain rewards, resulting in a greater pragmatic value. Similarly, a higher pragmatic value also leads to a lower expected free energy.

Planning and action selection are conceptualized as a process of minimizing expected free energy in active inference—that is, the agent seeking a policy that maximizes the sum of pragmatic value and epistemic value ([33]; [55]). Consequently, during behavioral selection, the agent not only seeks to maximize reward returns by exploiting known resources but also pursues exploring unknown information to reduce uncertainty. Expected free energy offers a principled solution to the exploration–exploitation dilemma in repeated games ([27]). Exploration seeking epistemic value and exploitation seeking pragmatic value are regarded as two equally important aspects of minimizing expected free energy. The exploration–exploitation trade-off transitions from a sequential decision-making problem in reinforcement learning to the optimization of a single objective function associated with expected free energy minimization. The choice to explore or exploit depends on current levels of uncertainty and the level of expected reward ([65]). It is worth emphasizing that the epistemic value term within expected free energy formally instantiates a mechanism for directed exploration, analogous to curiosity, driving the agent to autonomously and actively seek observations that reduce uncertainty about hidden states ([21]; [54]).

## 6. Why Active Inference? Advantages for Decision-Making in Repeated Games

Active inference, a theoretical framework grounded in generative models, commonly formulated as Partially Observable Markov Decision Processes (POMDP), offers both depth and breadth for understanding decision-making in repeated games. Its strength lies not only in its ability to unify multiple cognitive processes under the single principle of minimizing free energy, but also in its capacity to more closely capture the core essence of human social interaction. Future research could validate the effectiveness of active inference models through computational model comparisons and simulations. Furthermore, by fitting these models to behavioral game data and neuroimaging data, we can rigorously test the psychological and neural significance of their parameters, thereby bridging the gap between computation, behavior, and brain function.

### 6.1. The Free-Energy Principle: A Unified Framework for Cognitive Integration and Behavioral Optimization

The free-energy principle posits that any self-organizing system (such as the brain) within a changing environment must minimize its free energy to maintain the homeostasis necessary for survival ([24]). Free energy, serving as an upper bound on surprise, is a tractable measure of prediction error. The most fundamental theoretical advantage of the active inference framework lies precisely in this free-energy principle. It transcends the paradigm of traditional models that treat perception, learning, planning, and action selection as separate modules, offering instead a unified and biologically plausible computational framework. This framework conceptualizes all these cognitive processes as different manifestations of a single principle: free energy minimization ([22]). This characteristic gives it with exceptional explanatory power for modeling the complex decision-making processes inherent in repeated games. Within the context of repeated games, a player’s perception of their opponent (inferring their hidden states), immediate action selection (choosing to cooperate or defect), and learning from multi-round interactions (updating beliefs about the opponent’s behavioral patterns) are no longer treated as separate cognitive processes. Instead, they are integrated into a unified process of active inference under this single framework.

Conventional cognitive science research often presupposes different optimization objectives for distinct cognitive functions such as perception and action. For instance, perception optimizes for accuracy, while action optimizes for utility. In social decision-making, the accuracy of individuals’ inferences about another person’s thoughts and feelings is crucial, as it pertains to prediction, control, and decision outcomes ([79]). Researchers focus on how to optimize the accuracy of interpersonal perception within social interactions ([38]). At the action level, from a game theory perspective, the goal of an individual in a game is to continually optimize their behavior to approximate an optimal strategy, seeking to maximize their own utility ([8]). Faced with this perception and action that have different optimization targets, the active inference framework allows them to align around the same fundamental objective: minimizing the model’s prediction error—the minimization of free energy. In terms of perception, minimizing free energy involves updating the internal state of others’ thoughts and feelings to reduce the discrepancy between the “anticipated thoughts of others” and the “actually observed interactive cues,” ultimately enhancing the accuracy of interpersonal perception. In terms of action, minimizing free energy involves adjusting one’s own behavior to make the “actual outcome of the interaction” align more closely with the “preferred anticipated outcome,” thereby increasing behavioral utility.

Furthermore, regarding the core issue of the exploration–exploitation dilemma in repeated games, the free energy principle similarly offers an endogenous solution, enabling a dynamic balance in behavioral optimization. Traditional reinforcement learning models often depend on externally tuned parameters to regulate exploratory versus exploitative behaviors. For instance, in the ε-greedy strategy, ε represents the probability of exploration, while 1 − ε corresponds to exploitation, with ε commonly scheduled to decay over the learning process ([77]). The setting and adjustment of ε are generally guided by empirical or mathematical considerations rather than being grounded in a well-founded cognitive mechanism. In active inference, by contrast, this trade-off is internalized through the minimization of expected free energy, which reframes it as a unified optimization problem with a more principled cognitive basis ([21]). In the mathematical formulation of expected free energy, pragmatic value motivates exploitative behavior, whereas epistemic value promotes exploratory actions ([39]). Exploration and exploitation are integrated into a single objective function (surprise minimization), where action selection aims to maximize the combined epistemic and pragmatic value ([22]). Thus, compared to classical expected utility maximization, surprise minimization under the free energy principle does not contradict but extends it by incorporating epistemic value. This allows the model to adapt dynamically to the game context. In early interaction stages, when social uncertainty is high, epistemic value dominates, leading to pronounced exploratory behavior. As beliefs about the opponent become more precise, uncertainty decreases, the weight of epistemic value diminishes, and pragmatic value increasingly guides decision-making, resulting in a natural shift toward exploitation. This dynamic balance arises from the evolution of internal belief states rather than from external parameter tuning, significantly enhancing the model’s realism and explanatory power in describing human behavior in repeated games. Moreover, the free energy principle provides a compelling account of commonly observed behaviors, such as risk-taking in high-reward contexts and curiosity-driven exploration in the absence of extrinsic incentives ([65]). By incorporating the exploration–exploitation trade-off into the unified framework of the free energy principle, the active inference framework exhibits strong explanatory power in both behavioral optimization and cognitive integration.

### 6.2. Simulating the Nature of Social Interaction

The core challenge of social decision-making stems from social uncertainty, which is ubiquitous in the social world—not merely in repeated game scenarios. During social interactions, the thoughts and intentions of others are largely hidden, making it difficult to infer their behavior and its implications for us ([18]; [37]). Active inference, grounded in the framework of Partially Observable Markov Decision Processes (POMDP), naturally accounts for this uncertainty and unobservability into its models ([65]). Furthermore, active inference inherently incorporates the behavioral drive to resolve this uncertainty into its mathematical formulation. Extensive prior research has focused on exploring the influence of rewards and punishments on social decision-making, revealing that reward-related computations and neural circuits play a crucial role in guiding choices, similar to non-social decision-making ([61]). However, another equally important guiding factor has been overlooked: the motivation to reduce social uncertainty ([3]). In social interactions, people often engage in exploratory behaviors to reduce uncertainty; these actions do not confer direct rewards and may even involve risks and costs. The emergence of such behavior indicates that beyond the practical rewards brought by decision-making, understanding others’ thoughts and reducing social uncertainty themselves also has intrinsic value—a value distinct from external rewards ([44]). In social interactions, individuals possess a proactive desire to explore others’ thoughts and intentions, known as interpersonal curiosity ([82]). To satisfy interpersonal curiosity, individuals need to reduce uncertainty. The expected free energy of active inference incorporates epistemic value into behavioral choices. Epistemic value, also known as intrinsic value, mathematically corresponds to information gain or a reduction in uncertainty ([21]). According to the theory of expected free energy, an individual’s behavioral choices aim not only to maximize pragmatic value but are simultaneously driven by the maximization of intrinsic epistemic value. This computational mechanism provides a clear explanation for why individuals exhibit interpersonal curiosity during social interactions.

In social interactions, individuals need to infer others’ thoughts, perspectives, emotional states, and behavior patterns, and more. This involves both automatic inferences that form initial impressions and controlled inferences like perspective-taking ([17]; [34]). When modeling this mentalizing process in repeated game scenarios, the hierarchical model of active inference offers an elegant framework ([23]; [58]). Its Bayesian network representation is shown in Figure 4d. To facilitate understanding of the complex hierarchical model, Figure 4 illustrates the evolution of various active inference models, progressing from simple to complex. In the figure, circles represent variables, squares represent factors that modulate relationships, and arrows indicate dependencies between variables. Figure 4a depicts a generative model for static perception, involving only a single time point, analogous to standard Bayesian inference. Consistent with the earlier description of Bayes’ theorem, *s* represents the abstract hidden state, *o* denotes the observable outcome, *D* represents the prior belief about the hidden state *s*, and *A* is the likelihood function, specifying the probability of observing *o* given the state *s*. Figure 4b shows a generative model for dynamic perception. Compared to Figure 4a, it incorporates two or more time points and includes a state transition matrix *B*, which describes how the hidden state *s* evolves over time. Figure 4c builds upon Figure 4b by adding policy selection, representing dynamic perception with action choices. Here, *π* denotes a policy, where different policies correspond to different state transition matrices. *G* represents the expected free energy, and *C* represents the preferred outcomes. Figure 4d illustrates the hierarchical model, which comprises two levels of generative models. In this architecture, the hidden state *s*^2^ at the higher level provides the prior belief for the hidden state *s*^1^ at the lower level. Conversely, the posterior belief of the lower-level model is treated as an observation at a given time point for the higher-level model. The likelihood matrix *A*^2^ of the higher-level model mediates this bidirectional information flow between levels. This structure allows the higher-level model to update at a slower temporal scale compared to the lower-level model, which is why this framework is also referred to as the deep temporal model.

In repeated game scenarios, understanding the thoughts and goals underlying others’ actions typically requires observing a series of specific behaviors within the game. This is analogous to inferring a higher-level intention behind a shot (e.g., forehand or backhand) in tennis by observing the movement characteristics of various parts of the opponent’s body ([58]). Within the hierarchical active inference model, the agent uses the lower level of the hierarchy to process the specific choices in each round of the game, while the upper level characterizes the opponent’s higher-level, more stable attributes—such as intentions, strategies, or personality traits. For example, in multi-round prisoner’s dilemma games, the prior beliefs *D*^2^ in the upper-level model encode the automatically inferred initial impression of the opponent. The hidden states *s*^2^ encode the opponent’s different strategic modes (such as cooperative, deceptive, tit-for-tat, etc.). The opponent’s strategy may change slowly over time, and the agent gradually updates these high-level beliefs through perspective-taking and learning processes, which is consistent with the characteristics of the upper-level model. The high-level beliefs about the opponent’s strategy pattern, combined with the high-level likelihood matrix *A*^2^, generate predictions regarding the opponent’s specific behavior in the next round. These predictions serve as prior beliefs for the lower-level model, thereby guiding the agent’s own specific behavioral choices. The hierarchical model offers considerable flexibility, allowing it to be tailored to different game contexts. It enables the modeling of an individual’s high-order beliefs, making it possible to simulate mentalizing during social interactions and thereby providing a closer approximation to human social decision-making processes.

### 6.3. Future Directions: Model Simulation and Behavioral Fitting

The explanatory power of active inference is demonstrated not only at the theoretical level, such as through the free energy principle, but also in its provision of a computationally tractable and testable generative model that effectively bridges theory and data. Using simulation studies, researchers have shown that active inference models generate predictions about perception and behavior across domains such as action understanding, cognitive control, and sport anticipation that closely match empirically observed patterns ([29]; [57]; [58]). Other researchers have employed model fitting on behavioral data from cognitive processes such as approach-avoidance conflict, interoceptive inference, and directed exploration, extracting key model parameters that act as effective biomarkers for distinguishing between healthy individuals and those with psychopathology ([66], [67], [68]). Beyond computational psychiatry, studies that fit active inference models to behavioral data from sport anticipation tasks have likewise yielded parameters that effectively differentiate experts from novices ([30]). However, relatively few researchers have applied active inference models to the study of social cognition using either simulation or behavioral fitting methods. The theoretical advantages of the active inference framework for understanding social decision-making in repeated games likewise still require rigorous testing.

Model simulation and behavioral fitting are two complementary methodologies that form a progression from theoretical verification to empirical testing. Model simulation constructs models based on theory to generate simulated data. It is a deductive process conducted in an idealized environment, which allows researchers to test a theory’s internal consistency and generative capacity. Future research could define the state space (hidden states *s*, observations *o*), likelihood matrix *A*, and state transition matrix *B* for generative models based on specific game tasks. By adjusting parameters such as prior beliefs and the precision of expected free energy, researchers can observe whether the agent’s policy patterns align with empirically observed behavior. For instance, a hierarchical active inference model can be constructed on the basis of the Prisoner’s Dilemma game rules. By manipulating the priors of the higher-level model to simulate agents with different initial impressions of their opponent (e.g., expecting cooperation vs. defection), researchers could investigate whether the process of forming cooperation differs, thereby testing whether the model can reproduce empirically observed strategic patterns. Building upon the preliminary validation of a model’s effectiveness through simulation, behavioral fitting further applies this model to empirical behavioral data. Starting from real data, this approach either compares the model’s goodness of fit or estimates its parameters. As a form of model inversion, behavioral fitting tests the theory’s external validity and explanatory power. By collecting behavioral data (e.g., specific choices, reaction times, subjective reports) from participants in game experiments, researchers can compare the active inference model with classic computational models such as reinforcement learning. This comparison tests whether the active inference model can effectively explain human behavioral data and evaluates its goodness of fit. The parameters estimated from fitting can serve as computational biomarkers to distinguish between groups and to quantify individual differences in social decision-making. Future work may design experimental tasks that encompass different types of games to collect human behavioral data. Using the same active inference model, researchers could fit data across game types by adjusting specific structural parameters only as a function of the game types. If the model successfully explains human behavior across diverse game contexts, this would demonstrate its potential and generality as a unified theory of social decision-making. Furthermore, integrating neuroimaging data with computational model parameters will help reveal the neurocomputational mechanisms underlying social decision-making, thereby promoting the broader and deeper application of active inference in social cognitive neuroscience.

However, it must be acknowledged that applying active inference to the study of social decision-making presents several challenges. Active inference remains a predominantly theoretical framework, and its empirical application is still in early stages. Its high computational complexity and large number of parameters make the modeling process relatively challenging for researchers in psychology. Furthermore, existing guidelines for constructing active inference models are primarily based on relatively simple experimental tasks ([65]); transferring them to more complex repeated game is a formidable undertaking. Concurrently, researchers in the philosophy of science have questioned the free energy principle ([12]), arguing that it is overly abstract and general, and cannot replace research into the specific mechanisms underlying cognition and behavior. This serves as a crucial reminder that while we recognize its substantial explanatory power as a theoretical framework, equal emphasis must be placed on empirical research to validate and substantiate the concrete implementation of its computational models.

## 7. Conclusions

This review has systematically examined the potential of the active inference framework to illuminate the cognitive mechanisms underlying decision-making in repeated games. Unlike one-shot games, repeated games more closely resemble real-life social interactions and thus serve as a classic paradigm for studying social decision-making. The difficulty of directly observing others’ thoughts and motivations introduces social uncertainty, which significantly increases the complexity of decision-making in repeated games. Individuals must engage in real-time learning and strategic adjustments based on feedback across multiple rounds of interaction, a process that involves interconnected cognitive components such as inference, strategy generation, and learning. Faced with such complex cognitive processes, traditional reinforcement learning models, while adept at capturing experience-driven value updating, are limited in their ability to handle incomplete information and to infer opponents’ strategies. Bayesian inference can accurately describe belief updating and uncertainty management, yet traditional exact Bayesian methods often struggle to fully integrate with action policy selection. Against this backdrop, the active inference framework, through the free-energy principle, unifies perception, learning, planning, and action within a single generative model structure. Perception and learning are achieved by minimizing variational free energy, thereby balancing model complexity against accuracy. Planning and action selection are realized by minimizing expected free energy and thereby maximizing both epistemic and pragmatic value. Compared to other models, the active inference framework offers a powerful approach to tackling the critical exploration–exploitation dilemma inherent in repeated games. Formulated based on partially observable Markov decision processes (POMDPs), the framework naturally incorporates social uncertainty. Furthermore, its hierarchical structure affords a plausible representation of the mentalizing processes that are central to social interaction. Future research should further evaluate the framework’s empirical validity using complementary methods of model simulation and behavioral fitting. In summary, the active inference framework provides a unified and explanatory perspective for understanding human behavior in repeated games. It offers a rigorous methodology for uncovering the hidden dynamics of social decision-making and thus positions itself as a potent tool for advancing computational social psychology.

## Figures and Tables

**Figure 1 behavsci-15-01727-f001:**
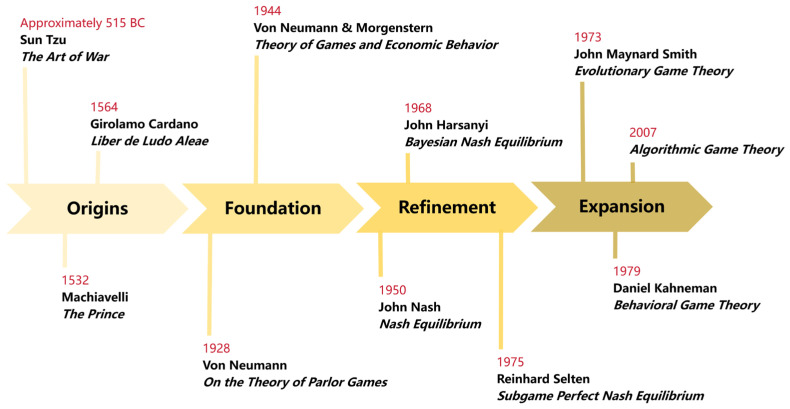
The history of game theory development.

**Figure 2 behavsci-15-01727-f002:**
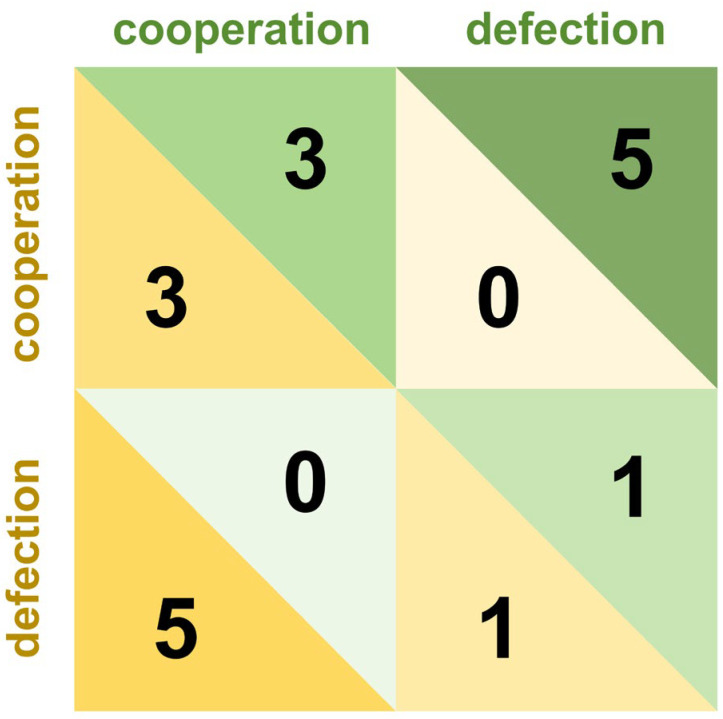
Decision and payoff matrix of the Prisoner’s dilemma game.

**Figure 3 behavsci-15-01727-f003:**
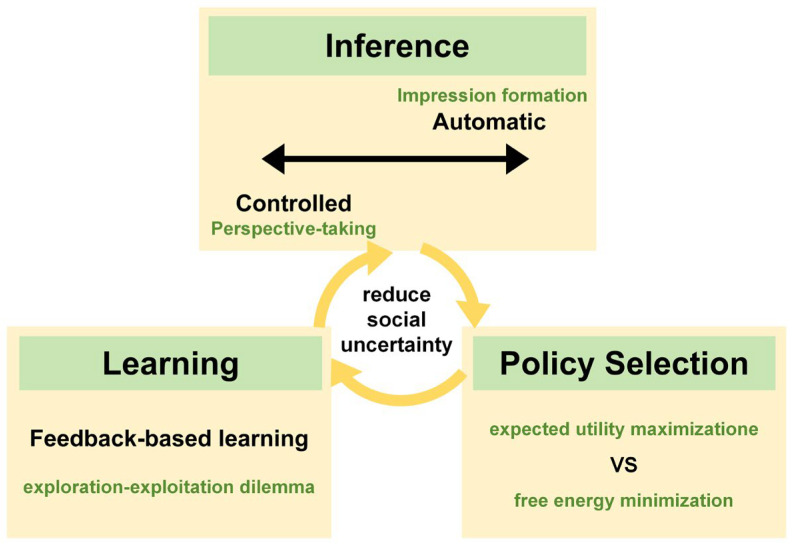
Decision-making process in repeated games.

**Figure 4 behavsci-15-01727-f004:**
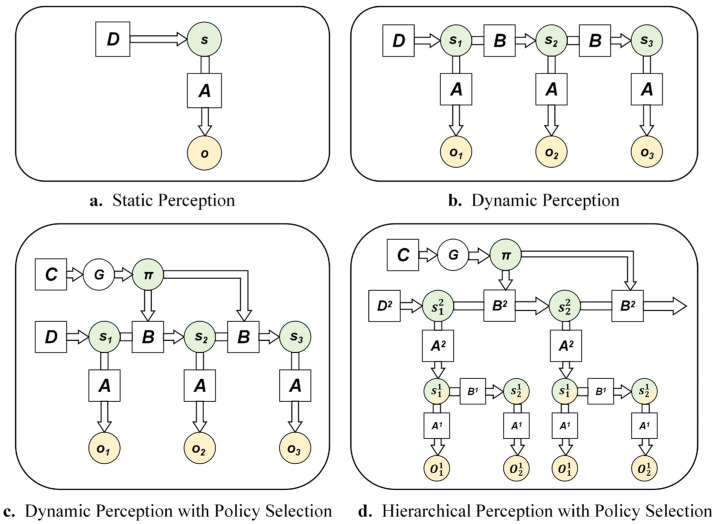
Bayesian network representation of various active inference models.

## Data Availability

No new data were created or analyzed in this study. Data sharing is not applicable to this article.

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
