# Peer review of "Decision-Making in Repeated Games: Insights from Active Inference"

_behavsci, 2025, doi:10.3390/bs15121727_

Round 1

Reviewer 1 Report

Comments and Suggestions for Authors

Moreover, the authors could present classification of different games.

Finally, the authors did not cite classics of the Game Theory and Nobel Prize winners in this research.

Discussions on the mathematics of games began long before the rise of modern, mathematical game theory. Cardano wrote on games of chance in Liber de ludo aleae (Book on Games of Chance), written around 1564.

The work of John von Neumann established game theory as its own independent field in the early-to-mid 20th century, with von Neumann publishing his paper On the Theory of Games of Strategy in 1928. Von Neumann's original proof used Brouwer's fixed-point theorem on continuous mappings into compact convex sets, which became a standard method in game theory and mathematical economics. Von Neumann's work in game theory culminated in his 1944 book Theory of Games and Economic Behavior, co-authored with Oskar Morgenstern.

The authors use proper methodology of a study.

The conclusions are valid and supported by sufficient data.  Meanwhile, any review of theory must start from classification of different approaches. Therefore, the authors should present it.

Before classification, the authors must present short history of the Game Theory, including fathers and most prominent scientists in the field. Then the authors must present the most prominent scholars in the field of Repeated Games.

The authors could to draw classification and present graphically research flow and findings.

The conclusions are consistent with the evidence and arguments presented and they address the main question posed.

The references appropriate. Meanwhile the authors could extend its including more classics and Nobel Prize winners.

Comments on the Quality of English Language

English language is good. In some places it could be better.

Reviewer 2 Report

Comments and Suggestions for Authors

This paper explores how Active Inference (often called the Free Energy Principle), a theory to describe how the brain works, can be used to understand decision-making processes in repeated games. Reviewing the traditional methods like reinforcement learning (or maybe evolutionary game theory), which assumes individuals always try to behave in such a way that the outcome maximizes the benefit (reward or payoff), the authors introduce a new view based on "Active Inference". In this perspective, players (or agents) choose actions to minimize variational free energy (which can be regarded as their expected prediction error). Essentially, the agents act to make their internal predictions match what they actually observe in an environment. They argue that this framework provides a normative account of strategic behavior in repeated interactions.

I find the approach described in this paper to be insightful and I think the paper provides a fresh way to study repeated games. It is well written and I believe it can be accepted for publication as is.
